# A Graphical Model-Based Representation for Classical AI Plans using Category Theory

**Angeline Aguinaldo[1, 2], William Regli**

[1]University of Maryland, Dept. of Computer Science,
[2]Johns Hopkins University Applied Physics Laboratory,
aaguinal@umd.edu, regli@umd.edu

## Abstract

Classical AI planners provide solutions to planning problems in the form of long and opaque text outputs. To aid in the generalization of planning solutions, it is necessary to have a rich and comprehensible representation for both human and computers beyond the current line-by-line text notation. In particular, it is desirable to encode the trace of literals throughout the plan to capture the dependencies between actions selected. The approach of this paper is to view the actions as maps between literals and the selected plan as a composition of those maps. The mathematical theory, called category theory, provides the relevant structures for capturing maps, their compositions, and maps between compositions. We employ this theory to propose an algorithm agnostic, model-based representation for domains, problems, and plans expressed in the commonly used planning description language, PDDL. This category theoretic representation is accompanied by a graphical syntax in addition to a linear notation, similar to algebraic expressions, that can be used to infer literals used at every step of the plan. This provides the appropriate constructive abstraction and facilitates comprehension for human operators. In this paper, we demonstrate this on a plan with the Blocksworld domain.

## Introduction

Plans generated by automated planners for real-world problems are often long and opaque. They are multi-line text files, where every line describes the action and the associated parameters for a given step. This is a simple and interpretable form for computers, but not very intuitive for humans to understand. It is often difficult to understand why the planner chose that sequence of actions without looking deeper into the domain description file and interpreting every action's cause and effect. And often the domain files can consist of tens or possibly hundreds of defined actions, making it incredibly difficult to parse.

One approach to improving comprehension is to make the domain description more intuitive. This can be done by using syntax highlighting or graph-based representations of action relationships. However, this requires that the user take the plan, compare it against the domain representation, and make effortful inferences about why the actions were selected. Another approach would be to look into the planner

and provide a representation about the search path and state space. This is useful for intuiting the efficiency of the planning algorithm; however, may not be relevant to an end user trying to understand why the plan was chosen. This motivates the cause for a representation of the plan in the context of its domain, and one that specifically highlights the provenance of literals from initial to goal state.

The widely-used convention for encoding artificial intelligence (AI) planning problems is the Problem Domain Definition Language (PDDL) (Ghallab et al. 1998). At the backbone of PDDL reasoning are logical propositions, or literals. These literals describe the preconditions and effects of an action. Provided a sequence of actions in a plan, it is likely that not all of an action's effects map to the next action's preconditions. This makes it difficult to understand which literals are informing each action because they may have been introduced many steps prior. This may lead the user to inquire about the purpose each action has in achieving the goal. Towards an explainable plan, it is appealing to have a graphical representation that traces the literals as they are used and generated by actions. This paper proposes using a representation, called string diagrams, that is based on the branch of mathematics called category theory. Category theory is an algebraic system that is attentive to functions and how they compose. This is a good representation for PDDL plans because they are compositions of actions in an ordered sequence. Validating compositionality between within and between plans is exactly the property to leverage when evaluating the transferability of skills to other plans.

In this paper, we provide introductions to PDDL, category theory, and string diagrams and discuss related representations for PDDL domains and plans. We then present an example visualization for the Blocksworld domain file and then describe how we encoded PDDL domains, problems, and plans into the string diagram representation. Lastly, we discuss notable observations and how this representation can be extended to support explainable planning.

## Background

### Planning Domain Definition Language (PDDL)

The prevalent classical planning language for describing domains and problems is the Problem Domain Definition Language (PDDL) (Ghallab et al. 1998). The schema adopted

by PDDL at inception was based on the language used by the Stanford Research Institute Problem Solver (STRIPS) (Fikes and Nilsson 1971). This requires a set of propositions, $F$, a set of operators with preconditions and effects, $O$, an initial state, $I$, and a goal state, $G$, and operates under the *closed world assumption*—all absent information is negative information. In other words, a STRIPS-based PDDL planning model can be defined as $P =< F, O, I, G >$. State models, $S(P)$, is a set of states, $s \in S(P)$, such that its elements are propositions. $A$ is the ground set of actions obtained from $O$. A transition function, $f : A \times S \to S$, maps between states according to the action applied. A corresponding cost is computed using a cost function, $\sigma : A \times S \to \mathbb{R}$. A plan, $\pi =< a_i, a_{i+1}, ...a_n >, a_i \in A$, is the sequence of actions that transition from the initial state, $s_0 = I$, to the goal state, $s_G = G$, according to the transition function, $f(a, s)$ and cost, $c(\pi) = \sum_{j=i}^{n} \sigma(a_j, s_j)$ (Geffner and Bonet 2013). There have been a number of solver heuristics and search algorithms developed that design cost functions according to soundness and optimality in order to identify plans efficiently (Ghallab, Nau, and Traverso 2004).

## Model-based Representations for PDDL

Explainable artificial intelligence (AI) planning (XAIP) is a subarea of research within the field of explainable AI (XAI) (Gunning 2017) whose goal is to relay to the user how and why a sequence of actions have been selected as a plan or policy (Fox, Long, and Magazzeni 2017). One approach to XAIP explanations is to use a representation that considers a plan in the context of the original domain model, i.e. *model-based representation*. A model-based representation for AI plans is one that relies only on the solution, domain, and problem model provided by the user which means it is agnostic to the method used to produce the plan and is typically more relevant to the end user than algorithm-based or planning-based representations (Chakraborti, Sreedharan, and Kambhampati 2020).

A common investigatory question that is asked of a model-based representation is *"Why is this action in the plan?"*. A good explanation for this query is one that shows how the goal depends on the chosen actions of the plan (Chakraborti, Sreedharan, and Kambhampati 2020). There are few model-based representations that provide such explanations for PDDL plans. A prominent contender is Dovetail (Magnaguagno, Pereira, and Meneguzzi 2016). Dovetail is a 2D graphical representation that uses jigsaw puzzle shapes an analogs for literal preconditions and effects. Every literal in the plan is assigned a vertical position, or row, in the visualization and every action in the plan is given a column. As a result, consecutive action pieces can fit together if their literals are compatible. This metaphor is intended to communicate the naturalness of the selected chain of actions. This is a creative view of the plan, however, it lacks the formal structure that would be used to provide information without visual inspection.

An example of mathematical model-based representations for PDDL are directed graphs. GIPO, Graphical Interface for Planning with Objects, (Simpson, Kitchin, and McCluskey 2007) and VisPlan (Glinský 2011) propose the use of di-

rected graphs to model PDDL plans. Graph representations are useful for encoding relationships between actions according to their ingoing and outgoing literals, but their mathematical structures do not inherently capture the order in which actions are executed which is necessary for explaining causal dependency of actions.

## String Diagrams and Category Theory

Eilenberg and MacLane (Eilenberg and MacLane 1945) introduced the concepts of category theory in their study of algebraic topology as a way to transfer theorems between algebra and topology. In doing so, they provided a mathematical language that lifts many mathematical and non-mathematical concepts to this notion of maps between entities and compositions of those maps. This abstraction has found its usefulness in modeling natural language (Coecke, Sadrzadeh, and Clark 2010), manufacturing processes (Breiner, Jones, and Subrahmanian 2019), database schema integration (Shinavier and Wisnesky 2019), biological protein structures (Spivak et al. 2011), and many other domains that require observing the interactions between entities, as opposed to the entities themselves. Likewise, these concepts provide a useful presentation of AI planning domains and plans because the resulting plans can be thought of as serial and parallel composition of maps between states.

To formally specify this representation, a mathematical structure must be defined. In category theory, the mathematical structure used is called a *category*. To define a category, $\mathbb{C}$, it must have a set of *objects* $\{A, B, C, ...\}$ and a set of *morphisms* $\{f, g, ...\}$ that map between objects. The map for a given morphism can be written as $f : A \to B$. The source object is called the *domain* and the target object is called the *codomain* (Spivak 2014). These objects and morphisms satisfy the following (MacLane 1971):

- For every object, there exists an *identity morphism*.

$$\forall A \in \mathbb{C}, \ id_A : A \to A \qquad (1)$$

- The *composition* operation, $\circ$, acts on morphisms. Morphisms are composable when the codomain of a morphism exactly equals the domain of another morphism.

$$f : A \to B, \quad g : B \to C$$
$$g \circ f : A \to C \qquad (2)$$

- The composition of morphisms is *associative*.

$$f : A \to B, \quad g : B \to C, \quad h : C \to D$$
$$(h \circ g) \circ f = h \circ (g \circ f) \qquad (3)$$

- Identity morphisms act as a *left* and *right unitor of composition*.

$$id_B \circ f = f = f \circ id_A \qquad (4)$$

These properties enforce consistent behavior for when more morphisms are composed together.

To model more complex maps, it is necessary that the structure support multiple objects in the domain and codomain. Additional mathematical structure (tensor product $\otimes$) can be added to the definition of a category to support this. This enhanced category definition is known as a *symmetric monoidal category* (Joyal and Street 1991), $\mathbb{M}$, and has the following additional properties:

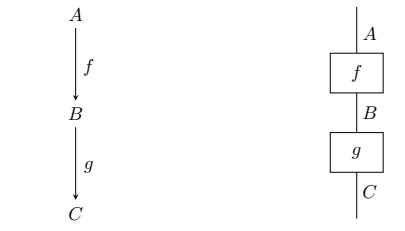

(a) Conventional syntax     (b) String diagram syntax

Figure 1: Two graphical representations for the composition of morphisms $f : A \to B$ and $g : B \to C$. In the case of (b), the lines can be called *strings* and these strings symbolize identity morphisms, i.e. $id_A$. A common shorthand, however, is to simply label the string according to the object as shown in (b).

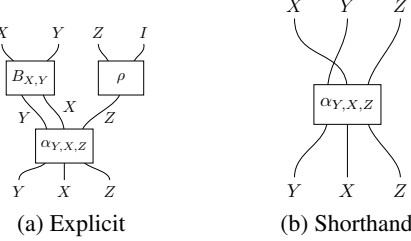

(a) Explicit     (b) Shorthand

Figure 2: A string diagram representation of the symmetric monoidal category properties shown in Equations 5-7. It is typical convention in string diagrams to replace rectangles symbolizing braids, such as $B_{X,Y}$ in (a), with crossing strings as seen in (b); and a tensor product with identity objects as just the non-identity objects.

- A *unit object*, $I \in \mathbb{M}$

- A map, called the *tensor product* $\otimes$, which is the product of $\mathbb{M}$ with itself, $\otimes : \mathbb{M} \times \mathbb{M} \to \mathbb{M}$.

- This tensor product is *associative*,

$$a_{X,Y,Z} : (X \otimes Y) \otimes Z \to X \otimes (Y \otimes Z) \quad (5)$$

- has *left* and *right unitor isomorphisms*,

$$\rho_l : I \otimes X \to X \qquad \rho_r : X \otimes I \to X \quad (6)$$

- and has a *braiding isomorphism* that is *symmetric* (Joyal and Street 1991)

$$B_{X,Y} : X \otimes Y \to Y \otimes X \quad (7)$$

These properties allow for both parallel and serial composition of processes. This also enables partial composition, where only a subset of domain strings of a morphism match the codomain strings of another morphism.

In addition to this structure, category theory provides a graphical syntax for illustrating maps and their compositions. Figure 1, shows the composition of $g \circ f : A \to C$ from Equation 2 in conventional syntax and a Poincare dual syntax. The Poincare dual syntax shows the morphisms $f$ and $g$ as rectangles and $A, B, C$ objects as lines. Figure 1b can be extended to support the symmetric monoidal category

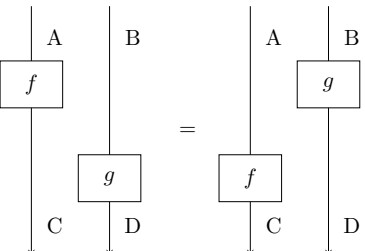

Figure 3: An example of deformation invariance in string diagrams. This feature allows the sliding of rectangles along strings in a planar fashion in accordance with the composition operator.

structure as seen in Figure 2, where multiple lines can pass through each rectangle. These diagrams are known as *string diagrams*.

In this syntax, horizontally adjacent strings represent the tensor product of those objects, i.e. $X \otimes Y$, which is another object in $\mathbb{M}$. The identity morphism of that object, i.e. $id_{X \otimes Y}$, can be used when composing, $\circ$, with other morphisms. An expression that describes the composition, $\circ$, and tensor product, $\otimes$, of morphisms is the linear notation that encodes its graphical representation. For example, the linear notation for Figure 2a can be seen in Equation 8.

$$
\begin{aligned}
(id_X \otimes id_Y \otimes id_Z \otimes id_I)\circ \\
(B_{X,Y} \otimes \rho)\circ \\
(id_Y \otimes id_X \otimes id_Z)\circ \\
(\alpha_{Y,X,Z})\circ \\
(id_Y \otimes id_X \otimes id_Z)
\end{aligned}
\quad (8)
$$

The morphisms and the order in which they are composed can be seen clearly from this notation.

A noteworthy consequence of the tensor product and its properties is the equivalence between strings diagrams under planar deformation (Selinger 2010). For example, we can consider the equivalence found in Equation 9 and its corresponding graphical syntax shown in Figure 3:

$$(id_C \otimes g) \circ (f \otimes id_B) = (f \otimes id_D) \circ (id_A \otimes g) \quad (9)$$

This gives the appearance of sliding rectangles past each other along strings which may be useful when seeking different sequences of compositions.

## String Diagram Notation for PDDL

In this section, we describe how we have chosen to translate key elements of PDDL domain, problem, and plan descriptions to mathematical structures from category theory. We assume that all morphisms and objects introduced belong to the same category. Table 1 summarizes this correspondence. Currently, this notation only supports the STRIPS requirement of PDDL 1.2 (Ghallab et al. 1998). Terms provided by other extensions such as typing, equality, conditional-effects are not handled.

## Domain File

The domain file provides the domain and codomain signatures for morphisms in our category. STRIPS-based domain files consists of the domain name, predicates, and actions.

**Actions**    Actions are operator data models that describe the parameters, preconditions, and effects of the given action, denoted by the `:action` token. Preconditions and effects values are conjunctions ($\wedge$) of logical predicates. A conjunction is distinguished by a pair of parentheses and the term `and`. The term `not` can prefix a predicate. When storing this negation, the parser prefixes the predicate name with the negation symbol, $\neg$. In the string diagram representation, actions are identified as morphisms, where each predicate in the precondition is a domain string and each predicate in the effect is a codomain string.

**Predicates**    Predicates are terms that serve as placeholders for data with logical states. Predicates are referenced in an action's preconditions and effects. They typically refer to aspects of the domain whose status will influence planning decisions. Some example predicates from the Blocksworld domain file include *(on ?x ?y)*, *(holding ?x)*, *(handempty)*. Predicates can be identified as strings when representing actions in the domain file as string diagrams. Provided a problem file and plan, parameters in the predicate signatures can be replaced with PDDL objects.

**Parameters**    Parameters for both the predicates and actions are prefixed with `?`. When parameters are used in predicate definitions, the characters following each `?` is used to distinguish that parameter from other parameters associated with that predicate. For example, in the predicate *(on ?x ?y)*, the characters *x* and *y* are not required symbols when using the predicate in the action conditions. In effect, these parameters are used solely to specify the arity of the predicate. These parameters will be substituted with objects specified by a PDDL plan. Parameters do not directly map to a structure in the string diagram representation.

## Problem File

The problem file informs the initial and goal strings in the string diagram. STRIPS-based problem files consist of the problem name, domain name, objects, an initial state, and a goal state.

**Objects**    Objects are symbols that are used to populate parameters, denoted by the `:objects` token. Objects do not directly map to a structure in the string diagram representation.

**Literals**    When predicate parameters are populated, according to a PDDL solver, they become literals, and are used as data for preconditions and effects in actions. In a string diagram representation, every uniquely parameterized predicate is treated as a string. For example, if two predicates *(ontable A)* and *(ontable B)* are constructed according to a

| PDDL File | Description | Category Theory |
|---|---|---|
| Domain | Actions | Morphisms |
| | Predicates | Objects |
| | Parameters | – |
| Problem | Objects | – |
| | Literals | Objects |
| | Initial State | Morphism |
| | Goal State | Morphism |

Table 1: Correspondence between category theory structures and description components found in PDDL domain and problem files

PDDL plan, they are unique strings. Notice that this also implies that negated versions of literals are considered distinct from their positive selves. For example, *(ontable A)* and *(not (ontable A))* are distinct strings and their relationship by logical negation is not encoded.

**Initial State**    The initial state, denoted by the `:init` token in the problem file, is a conjunction of literal assumptions that are true at the beginning of the planning problem. All literals in the initial state are represented by their own strings. These strings are tensored, $\otimes$, together and serve as the first morphism in the chain of compositions. In a later step, additional literals may be tensored with the initial state. This happens when an action in the plan makes an assumption about the initial state that is not explicitly stated in the problem definition.

**Goal State**    The goal state, denoted by the `:goal` token, is a list of literals that *must* be true at the end of the planning problem. All literals in the goal state are represented by their own strings. These strings are $\otimes$ together and serve as the last morphism in the chain of compositions. In a later step, additional literals may be tensored with the goal state if an action in the plan has an effect that is not explicitly required by the goal definition.

## Plan

After the sequence of parameterized actions have been converted into morphisms, we chain, or compose, these morphisms to generate a fully connected diagram. A naive chaining algorithm was implemented to infer valid tensor products and compositions in scenarios where morphisms can only partially compose. The primary goal of this algorithm is to enforce the compatibility of domains and codomains of preceding and subsequent morphisms during composition.

In this algorithm, there exists a notion of horizontal *slices* where each slice is a list of morphisms, including identities and braids, that will be tensored together from left to right. It is essential that the domain and codomain of the slices match in order to permit a valid composition.

There are four main steps to the algorithm:

1. *Backward pass* to weave input strings of each morphism, from goal state to initial state (bottom-up).

2. *Forward pass* to weave output strings of each morphism, from initial state to goal state (top-down).

3. *Add braids* to add braids, or string swaps, in case the order of the tensor product is not compatible for composition.

4. *Compose* to chain the blocks from top-down. This constructs the string diagram.

A byproduct of this chaining algorithm is the propagation of literals upstream and downstream which exposes those literals that are implicitly instantiated according to the plan.

## Example

A program written in the Julia programming language[1] was developed in order to automatically generate string diagrams. This program parses the PDDL domain, problem, and solution files and encodes its elements into the symmetric monoidal category constructs, which are JSON-serializable. Catlab[2], a Julia-based category theory library, was leveraged for its constructors. Currently, this program only supports PDDL files formatted using the STRIPS requirements.

### Blocksworld domain

The Blocksworld[3] domain describes a scenario where there are cube-shaped objects, called *blocks*, on a table, denoted by the predicate (*ontable ?x*). The objective is to stack the blocks according to the stacking configuration described by the goal. Only one block can fit on top of another block, (*on ?x ?y*), which implies a block cannot be stacked on more than one block simulataneously. If a block is not underneath another block, it is considered clear, (*clear ?x*). A hand is used to manipulate the configuration of the blocks, and can be empty (*handempty*) or holding a block (*holding ?x*).

The domain consists of four actions: pick up, put down, stack, unstack.

- `pick-up`: This action expects that a given block is clear, on the table, and the hand is empty. This action changes the state of the world such that the block is not on the table, the block is not clear, the hand is not empty, and the hand is holding the block.

- `put-down`: This action expects that the hand is holding a block. This action changes the state of the world such that the hand is not holding the block, the hand is empty, the block is clear, and the block is on the table.

- `stack`: This action expects that the hand is holding a block (`?x`) and that the other block (`?y`) is clear. This action changes the state of the world such that the hand is not holding the block (`?x`), the other block (`?y`) is not clear, the hand is empty, and the block (`?x`) is on top of the other block (`?y`).

- `unstack`: This action expects that the block (`?x`) is on top of another block (`?y`), that the block (`?x`) is clear, and the hand is empty. This action changes the state of the world such that the hand is holding the block (`?x`), the other block (`?y`) is clear, the block (`?x`) is not clear, the hand is not empty, and the block (`?x`) is not on top of the other block (`?y`).

The problem in this example initialized three blocks as objects, (*a, b, c*), and stated that all the blocks are clear and on the table, and that the hand is empty. The goal was to have block *c* on top of *b*, and block *b* on top of *a*.

```
(define (problem BLOCKS-3-0)
    (:domain BLOCKS)
    (:objects a b c)
    (:init (clear c) (clear a)
           (clear b) (ontable c)
           (ontable a) (ontable b)
           (handempty))
    (:goal (AND (on c b) (on b a))))
```

The PDDL4J toolkit[4] (Pellier and Fiorino 2018) was executed to determine the actions and parameters needed to transition the world from the initial state to the goal state. The planner output was parsed to extract the plan and ignore other outputs, such as cost and runtime. The parsed plan for this example can be seen below.

```
pick-up b
stack b a
pick-up c
stack c b
```

The individual steps in the plan can be understood as morphisms, and likewise can be represented as individual string diagrams. An example is shown in Figure 5. The diagrammatic description clearly depicts how the parameters populate the preconditions and effects to form literals; therefore, intuitively conveying the requirements of each action in the context of the problem. The fully-composed string diagram containing all the steps of the Blocksworld plan can be seen in Figure 7. The top of the diagram shows strings representing the literals of the initial state and assumptions about the world that were introduced by the plan. The bottom of the diagram shows strings representing the literals of the goal state as well as the literals produced from the plan. The diagram is read from top to bottom and the actions of the plan appear as labeled rectangles. Table 2 shows a summary of all the satisfied literals at every step in the plan. This information can be derived from the linear notation that encodes the graphical representation.

## Discussion

In this paper, we presented a model-based representation that describes a PDDL plan in the context of the domain description. From the string diagram depiction of the plan, we can clearly see what predicates were initialized, how they

---

[1]https://julialang.org/

[2]https://github.com/epatters/Catlab.jl

[3]https://github.com/pellierd/pddl4j/tree/master/pddl/blocksworld

[4]The default configurations of PDDL4J toolkit uses the heuristic search planner (HSP) (Bonet and Geffner 2001) and the FF heuristic (Hoffmann and Nebel 2001).

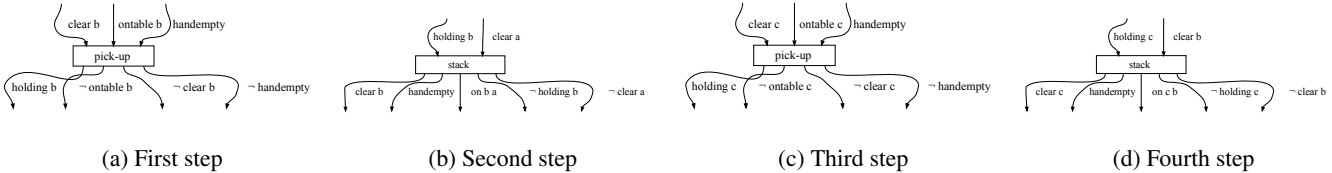

|  | (a) First step | (b) Second step | (c) Third step | (d) Fourth step |

Figure 4: These are the morphisms corresponding to steps proposed by a PDDL solver for the Blocksworld domain and problem described in the *Examples* section. The diagrammatic description of the steps clearly depicts how the parameters populate the pre- and post-conditions to form literals; therefore, intuitively conveying the requirements of each action in the context of the problem.

| State | Satisfied Literals | Action |
|---|---|---|
| **Initial** | **clear c**, **ontable c**, **clear b**, **ontable b**, **handempty**, **clear a**, **ontable a** | |
| **Step 1** | clear c, ontable c, **clear b**, **ontable b**, **handempty**, clear a, ontable a | pick-up |
| **Step 2** | clear c, ontable c, **holding b**, **clear a**, ¬ontable b,¬clear b, ¬handempty, ontable a | stack |
| **Step 3** | clear b,  **clear c**, **ontable c**, **handempty**, on b a, ¬holding b, ¬clear a, ¬ontable b, ¬clear b, ¬handempty, ontable a | pick-up |
| **Step 4** | **holding c**, **clear b**, ¬ontable c, ¬clear c, ¬handempty, on b a, ¬holding b, ¬clear a, ¬ontable b, ¬clear b, ¬handempty, ontable a | stack |
| **Goal** | clear c, handempty, **on c b**, ¬holding c, ¬clear b, ¬ontable c¬clear c, ¬handempty, **on b a**, ¬holding b, ¬clear a, ¬ontable b, ¬clear b, ¬handempty, ontable a | |

Table 2: The satisfied literals at every step of the plan can be concluded from the linear notation that encodes the string diagram representation shown in Figure 7. You can check this by reading the diagram from left to right at varying horizontal positions and noting the strings you encounter. The explicitly required literals are highlighted in **bold text**. The effects for each action are included as *Satisfied Literals* in the next row.

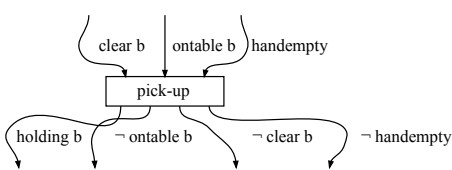

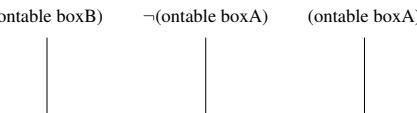

Figure 6: Strings for literals from the Blocksworld domain. The linear notation for the tensor product of these strings is $id_{(\text{ontable boxB})} \otimes id_{\neg(\text{ontable boxA})} \otimes id_{(\text{ontable boxA})}$.

Figure 5: This an example morphism corresponding to the first action chosen by a PDDL solver for the Blocksworld domain. The linear notation for this action is $pick\_up : (id_{(\text{clear b})} \otimes id_{(\text{ontable b})} \otimes id_{(\text{handempty})}) \rightarrow (id_{(\text{holding b})} \otimes id_{\neg(\text{ontable b})} \otimes id_{\neg(\text{clear b})} \otimes id_{\neg(\text{handempty})})$.

were used, and what predicates are present at any given point in the plan. Some notable observations that explicitly answer the question of *"Why is this action in the plan?"* or provide feedback to the domain designer are listed here:

- Following Figure 7 from top to bottom, we can observe that the first *stack* relies on *(holding b)* which is an effect from *pick-up*. This implies that *stack* depends on *pick-up* in order to execute.

- Additionally, we can observe when each literal in the initial state is used to inform the action. For example, the

*(clear c)* and *(ontable c)* literals do not get referenced until the second *pick-up* action is called.

- From the **Initial** row of Table 2, we can see that no new literals were introduced as implicit assumptions according to the given plan. This is evidenced by the lack of unbolded literals.

- Additionally, we notice the initialized literal *(ontable a)* was never used as a precondition to the actions of the plan. This may prompt the user to consider whether that particular assumption was necessary or whether this was expected behavior.

This representation shows the implicit changes occurring in the world that are not clearly evidenced by the simple line-by-line description of the plan. This also exposes the unanticipated literals that could result in errors when operating under the closed world assumption.

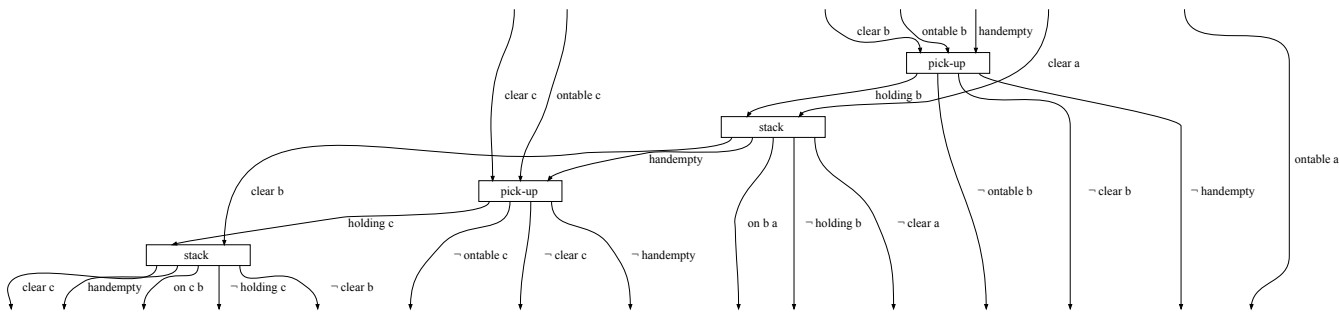

Figure 7: String diagram for the Blocksworld domain and problem. The diagram is read top-down. The curved lines in the illustration, i.e. strings, represent literals that are initialized or constructed according to the PDDL plan. The rectangles represent the actions from the domain that are referenced in the plan. The order in which the actions appear in the plan can be seen by following the strings from the top to the bottom and noting the order of the rectangles you encounter along the way. The entire diagram represents a map from initial state to goal state.

## Limitations

The current encoding scheme presents some limitations. As aforementioned, the PDDL extensions supported are restricted. We currently do not have a way to visualize notation such as quantifiers, equalities, and other extensions. We are also unable to encode relationships between positive and negated version of the same literal, which are currently treated as independent information under the closed world assumption. Lastly, the visualization does not scale effectively to long plans with many actions. To handle this, it may be possible to design a heuristic to detect repeated patterns in actions so that the plan can be grouped into subtasks.

## Benefits of category theory structure

Recall that this representation hinges on structures defined in category theory, such as morphisms and objects. Morphisms are made by the actions described in PDDL domain files. These morphisms can be seen as unit string diagrams that can be composed, as shown in Figure 5. This representation also provides the user with additional context for the actions, such as how the parameters populate the preconditions and effects. For example, in Figure 5, the step *pick-up b* is clearly depicted as having *(clear b)*, *(ontable b)*, *(handempty)* as pre-conditions and *(holding b)*, *(not (ontable b))*, *(not (clear b))*, *(not (handempty))* as effects, which is not easily inferred by the text description of the plan.

We also introduced that fact that these morphisms can be vertically, ∘, and horizontally, ⊗, composed together. The linear expression translates to the layout of the rectangles and strings in the string diagrams. Evidently, the layout is particularly useful for conveying a sense of order and time in the plan. Another key aspect of this is that the string diagram representation is deformation invariant (Selinger 2010), which means that sliding rectangles along strings is similar to rearranging terms of an algebraic equation. This implies that alternate but valid plans can be observed by re-ordering actions, i.e. rectangles. Notably, this structure is not limited to PDDL, but can be applied to any declarative domain language adhering to a functional paradigm.

## Future Work

The graphical representation and formal linear notation provide opportunities for explanatory insights and intuitive visualizations of PDDL plans. For visualization, it is easy to imagine interactions such as highlighting the strings of a particular literal in order to witness its path through the plan, or sliding rectangles along strings to view alternative plans. In addition, it may be interesting to scale the length of the strings or the height of the rectangles according to some solver metadata, such as cost, or a real-world parameter, such as time to execute. This notation has the potential of being incorporated in Web Planner (Magnaguagno et al. 2020) or PDDL Editor (Muise 2016) alongside Dovetail (Magnaguagno, Pereira, and Meneguzzi 2016) to provide another perspective to the PDDL plan.

## Conclusion

This representation has the primary benefit of providing a causal explanation of a PDDL plan with rigorous mathematical structure within the context of the domain and the problem. In particular, the diagrammatic syntax identifies rectangles as actions and curvy lines, or strings, as literals. The placement of the rectangles and strings relay a temporal progression, read from top to bottom, which allows user to sense the use of literals as the plan progresses. This can lead to insights about action dependencies and domain design improvements. Extensions of this work include adding interactivity to the graphical representation and manipulating layout and shapes in the diagram according solver metadata.

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
