# OpenReview forum: "A Graphical Model-Based Representation for Classical AI Plans using Category Theory"
_icaps-conference.org/ICAPS/2021/Workshop/XAIP — XAIP 2021_

### Official Review · AnonReviewer2 · 2021-07-04
**A Graphical Model-Based Representation for PDDL Plans using Category Theory**

**Rating:** 7
**Confidence:** 3

**Review:**

# A Graphical Model-Based Representation for PDDL Plans using Category Theory

## Summary
The paper presents a formal graphical output for plans obtained from classical planners.
The paper starts describing how plans are hard to follow based on the signature of actions, as the predicate relations described in the domain are lost.
The idea of model-based representations aiming PDDL is not new, as several papers with related work are described in the background.
The goal of the paper is to provide a graphical output with a formal mathematical structure that better explains the causal dependency of actions.
The formal mathematical structure is provided by string diagrams.
The string diagram notation for PDDL is limited to STRIPS, with each supported PDDL element described.
A single and small problem from the Blocksworld domain is presented as a complete example.
The paper finishes with a discussion about the example, limitations and benefits of the current solution and future work.


## Evaluation
The paper is good and fits XAIP, the solution looks well formalized with a clear example.
Small mistakes in the text can easily be corrected and the paper accepted.

**pros**
- Well written with clear notation and example.
- Fits XAIP and cites related work.
- Can be replicated based on the paper description.
**cons**
- Single example does not show how complex real-world output may become.
- Sometimes uses PDDL to refer to plans and planners, while PDDL is just one input language and the solution proposed could work for many.
- Some Figures are not referenced in the text and could be forgotten by a reader that follows text flow.

### Questions and notes about the text
**Why have PDDL in the title when this could support multiple action-based languages?**
PDDL, as far as I know, does not describe a standard output format, and the provided solution could be supported by any action language that matches the supported STRIPS.
The authors know this, as it is stated at the end of the Benefits section.
I believe the title could be improved by removing "PDDL", otherwise people using other planning languages, such as HTN planners, may never read about this solution.

The Abstract describes line-by-line plans as not comprehensible enough by computers, which is not true, otherwise it would be impossible to output any model from it.
The goal of the paper is to make things more comprehensible to humans with a visual aid.

The limitation of Dovetail requiring visual inspection seems also applicable to this work, as Web-Planner (tool behind Dovetail) is able to detail why a user-provided plan fails with a textual output.
It implies here that the proposed solution could solve this problem without visual inspection, but the notation seems very different/disconnected from the graphical output, at least in the current state.
Perhaps the authors want to say that it is easier to find a loose node in the resulting graph, which may not be true for complex plans.

**Being able to see both serial and parallel plans is quite interesting, could all parallel plans be visualized at the same time based on the provided plan used by the output tool?**
I think some plans would reorder too many actions and break the top to bottom ordering required to guide humans.
If this is a limitation, it should be described.

Typing and equality requirements could be "downgraded" to common predicates, such as (type obj) and (equal o o), and supported.
With that said I would simply ignore static predicates to simplify graphical and textual output.
The chosen example contains only fluent predicates and does not make clear how complex the graph can get when considering every precondition.

The order in which the domain file elements are described is weird, as Predicates and Parameters should come before the Actions.
The need for a plan in the Predicates description seems too focused on the final plan output, one could generate a single action with just a sequence of objects to replace the parameters.
An example is that one could generate all the string diagrams for each ground action, no planner or plan required.

The parameters are part of actions in PDDL, while the text implies they are separate elements, perhaps using free variables would be better.
The Parameters subsection says that parameters/free variables are only used to describe arity, which is only true when describing the predicate signatures.
The Action parameter names are actually meaningful in the grounding process when selecting which variable is replaced by which object.
The lack of ground parameters in the string notation seems odd, as it would add value to the output, "stack a b" is more comprehensible than "stack".

Objects are not only used to populate parameters, they describe every element that is described by the predicates.
Even if an object cannot be used as a parameters, it should be described if it is part of the initial/goal states or domain.

In Literals we see "PDDL solver" and "PDDL plan", perhaps "grounding scheme/procedure" and "plan" are better terms, could even cite a paper such as
"Concise finite-domain representations for PDDL planning tasks?" from Malte Helmert to make clear how parameters are replaced by objects.

The example describes a Julia program, if this software becomes public available, please add a link to the final paper.
**I would like more details about who decides the placement of elements in your solution, as only left-right and top-bottom constraints are imposed, is it all decided by Catlab?**
All actions have an unnecessary overlapping out edge, not sure if this adds meaning or just an artifact from the program.

Figures 4(a) and 5 are the same, while Figures 4 and 6 are loose, no part from the text references them.
This should be easy to fix, as the caption from Figure 5 is the actual text explaining details of Figure 4.

Not sure if "heuristic" is the the appropriate term in the Limitations section, perhaps procedure or function explains better how subplans/subtasks could be replaced by a single element.


## Small Improvements and Typos
- There is a loose "A" in the Background PDDL description, perhaps this is the ground set of actions obtained from O, this should be clarified.
- Use conditional-effects instead of condition-effects to match the requirement name.
- Initial and Goal state subsections are missing "the" before the keywords.
- Prefer executed instead of run
- Discussion: there is no "Start" row in Table 2, perhaps you mean "Initial".
- Double "was was" in the Discussion
- Figure 4 caption: missing "the" before "Examples section"
- "Figures 5" in Benefits should be Figure or add a missing second Figure number.
- Missing "to" in Conclusion "... according TO solver meta data".

---

### Official Review · AnonReviewer1 · 2021-07-05
**Plan visualization with formal foundations, contribution not so clear**

**Rating:** 7
**Confidence:** 4

**Review:**

The paper argues in favor of importing string diagrams from category theory into XAIP for plan visualization. Plan visualization is an enabling technique for plan inspection, debugging, and domain-engineering tools in general. Thus, the topic clearly is relevant to the XAIP community. Currently, there is not too much work done in this direction, so it would be good to see something happen. The approach taken in the manuscript proposes a graphical visualization with a formal semantics, which is a first step.

When it comes to the contribution and to results, the manuscript is very preliminary and sketchy. At various places, the approach gets compared to simply presenting the plan as a list of actions. This, of course, is a too simplistic baseline. Instead, the authors should compare their approach to other approaches to plan visualization (which get mentioned in the related work section, e.g., dovetail). Also, the additional value of using the category-theoretic semantics is not well explained. There is a shallow hint in the discussion section mentioning additional reasoning capabilities, but this aspect stays unclear. Moreover, as the authors also mention, it is obvious that the understandability of the diagrams will not scale well as plans become more complex. This is a weakness shared with other approaches. Empirical studies on the understandability of string diagrams were not conducted and not even mentioned as future work. In sum, from the current manuscript it does not become clear which problem the category-theoretic import really solves. If one were more critical, one could say it's just introducing a not well-known (among most XAIP researchers, I assume) formalism just to present actions along with their preconditions and effects in a graphical manner (in an quite obvious way, really), without actually improving state-of-the-art of plan visualization.

Despite these weaknesses, I think the manuscript works as a XAIP workshop paper to be discussed, and maybe it leads to some interesting results in the end.

---

### Meta-Review · Area_Chairs · 2021-07-07

**Recommendation:** Accept
**Confidence:** 4

**Metareview:**

Thank you very much for your submission.

Reviewers agree that the proposed graphical visualization with formal semantics may have potential, but the formalization and the clarification of advantages over other approaches need improvement.

The main discussion points are:

1. restriction to PDDL
2. possibility to visualize both serial and parallel plans
3. comparison to other plan visualization tools and advantages over them
4. additional value of using the category-theoretic semantics

We hope the reviews will be helpful. Please consider the comments for the camera-ready version. We look forward to your presentation!

---

### Decision · Program_Chairs · 2021-07-08

Accept